# Are We Identifying Malnutrition in Hospitalized Patients with Hematologic Malignancies? Results from a Quality Clinical Audit

**DOI:** 10.3390/diseases10030040

**Published:** 2022-07-04

**Authors:** Eftychia Kanioura, Ioannis-Georgios Tzanninis, Kalliopi-Anna Poulia, Aliki Stamou, Athanasios Liaskas, Dimitrios Politis, Athina Kaoura, Georgios Garefalakis, Nora Athina Viniou, Panagiotis Diamantopoulos

**Affiliations:** 1Hematology Unit, First Department of Internal Medicine, Laikon General Hospital, National and Kapodistrian University of Athens, Ag. Thoma 17, 11527 Athens, Greece; ekanioura@hotmail.com (E.K.); jg2911992@gmail.com (I.-G.T.); aliceastam@gmail.com (A.S.); ath.liaskas@gmail.com (A.L.); dpolitis92@yahoo.gr (D.P.); athina_1992@yahoo.gr (A.K.); garefalakisg@gmail.com (G.G.); noravi@med.uoa.gr (N.A.V.); 2Laboratory of Dietetics and Quality of Life, Department of Food Science & Human Nutrition, School of Food and Nutritional Sciences, Agricultural University of Athens, 75 Iera Odos, 11855 Athens, Greece; lpoulia@gmail.com

**Keywords:** disease-related malnutrition, audit, hematological malignancies, nutrition management

## Abstract

Disease-related malnutrition (DRM) is highly prevalent among patients with hematologic malignancies. The aim of the present study was to evaluate the prevalence of DRM in hospitalized patients with hematologic malignancies and investigate the level of awareness of DRM among the medical team treating this group of patients. A cross sectional quality clinical audit took place in two hematology units of a tertiary university hospital. Inpatients were screened within 48 h of their admission using the Malnutrition Universal Screening Tool (MUST) to identify their nutritional risk, and they were reassessed to identify the implemented interventions during their hospitalization. One hundred eighty-five patients were included in the audit analysis. On admission, 37.3% of the audited population was identified as being at high risk of malnutrition according to the MUST score. Forty-nine (26.5%) patients reported reduced food intake during the past 5 days, while four (2.2%) reported no food intake. During the hospitalization, only five patients (2.7%) received nutritional support, as indicated. Low levels of awareness of the early detection and treatment of DMS were found. Moreover, the prevalence of DRM and low nutritional intake was reported to be low. Measures to increase awareness of DMR in the medical team and better coordination of the nutrition support teams is vital to ensure better management and early nutrition intervention in hematological patients.

## 1. Introduction

Disease outcomes of patients suffering from hematologic malignancies have progressively improved [1]. Novel therapies and improvement in patients’ care have a tremendous impact on survival, but despite the advances in cancer research, various studies have indicated high prevalence of nutritional deficiencies and malnutrition among them [2,3]. Malnutrition can be defined as “a state resulting from a lack of intake or uptake of nutrition that leads to altered body composition (decreased fat free mass) and body cell mass, leading to diminished physical and mental function and impaired clinical outcome from disease” [4], whereas disease-related malnutrition (DRM) refers to a complex syndrome that combines the detrimental effects of insufficient nutritional intake and the disease-related systemic inflammatory response [4,5,6]. DRM in hospitalized adult patients has been connected with increased morbidity, lower performance status, increased length of stay (LoS), and increased mortality rates [7].

DRM in hematologic patients is often attributed to poor food intake due to primary anorexia or secondary causes, such as uncontrolled pain and chemotherapy complications [8]. Common therapy-related adverse effects are related to anorexia and odynophagia and include recurrent infections, oral mucositis, xerostomia, nausea, and vomiting. Additionally, hematologic malignancies are conditions of systemic inflammation with increased energy requirements. Poor nutritional intake can lead to protein catabolism and a negative energy balance, which leads to involutional weight loss and cachexia [8,9,10]. DRM attenuates the immune system response and impedes infection control and wound healing, leading to a slower recovery and poor clinical outcomes [11,12]. It has been shown that mortality rates are higher in malnourished patients admitted to general wards of a hospital (12.4% vs. 4.7% of well-nourished), LOS is higher (16.7–24.5 days vs. 10.1–11.7 of well-nourished), and health care cost can increase up to 308.9% [13,14]. These detrimental effects have a significant negative impact on the overall cost of treatment. According to a recent systematic review of the cost analysis nutritional support in medical inpatients, the provision of nutritional support in malnourished inpatients resulted in savings of 2818 USD per patient in a time period of 6 months [15].

Considering the negative impact on the patient’s prognosis and the financial burden of DRM, it is of paramount importance that organized strategies should be implemented. Nutritional screening should be performed on admission, followed by nutritional assessment for all patients found to be at risk of malnutrition, and nutrition interventions and frequent monitoring should take place during the patients’ hospital stay [9,16]. 

Clinical audits have been used to study the level of performance in the management of different clinical conditions. To date, there have been no studies focusing on the standards of care and nutritional support of patients with hematologic malignancies [10,17]. The present study was a cross sectional clinical audit on hospitalized patients with hematologic malignancies aiming to investigate the prevalence of DRM using the Malnutrition Universal Screening Tool (MUST), the level of awareness among the medical team treating this group of patients, the number of referrals to the dietitians, and the percentage of the patients at risk of malnutrition who received any type of nutritional intervention. Moreover, investigating the possible effect of DRM on hospital infections, LoS, and mortality was also a secondary aim of the study.

## 2. Materials and Methods

### 2.1. Study Design and Sampling

The present study was a cross sectional quality clinical audit in two hematology units of a tertiary public university hospital in Athens, Greece. The overall capacity of these units is 70 beds, and the average admissions per month is 100. Hospitalized patients were initially screened for DRM within 48 h of their admission to identify the risk of DRM. Then, if they were at risk of DRM, they were reassessed to identify whether any interventions were implemented. Inclusion criteria were a diagnosis of hematologic malignancy, and an expected LOS longer than 3 days. Exclusion criteria consisted of a delay in reassessment and the patient’s refusal to participate. The data collection took place from May 2018 to December 2018.

### 2.2. Data Collection and Measurements

Data were collected via a questionnaire that included the initial and the follow-up assessments. In the initial part, information about general demographic data, the duration of hospitalization, the diagnosis of the patients, nutritional risk screening by the MUST [16], prescription of hospital diet, nutrition intake of food other than the food provided in the hospital, and the provision of nutritional support were collected. The MUST tool is a screening tool introduced by the British Association of Enteral and Parenteral Nutrition (BAPEN) designed to identify patients at risk of malnutrition [18]. It has been validated in the clinical environment, in outpatient clinics, general practice, the community, and in care homes [18]. MUST score is calculated by the evaluation of recent unintentional weight loss, Body Mass Index (BMI), and the severity of disease, and the total score ranges from 0 to 2, where 0 corresponds to low risk, 1 to medium risk that needs monitoring, and 2 to high risk that necessitates action to treat [18]. 

Sources of information were the patients themselves, the patients’ environment in cases where communication with the patient was not feasible, the medical records, the medical and nursing staff, and the dieticians of the hospital. The performance status of the patients was evaluated using the Eastern Cooperative Oncology Group (ECOG) scale, a widely used method to assess the functional status of a patient. It classifies patients based on a 5-point scale, with 0 corresponding to a fully functional status and 5 to a very low performance status and death [19]. Data from the medical history, the diagnosis, the type of treatment (chemotherapy and/or radiation), and symptoms affecting nutritional intake, i.e., dysphagia, odynophagia, nausea, and signs and symptoms of concurrent mucositis, were also recorded. 

Anthropometric measurements were conducted for all patients and included measurements of height in meters, weight in kilograms (that was measured using the same scales with minimal clothing), and the usual reported weight, while any unintentional weight loss in the six previous months was also recorded. The BMI was calculated as weight in kg/(height in m)^2^. For patients who were unable to stand, the weight was calculated by calf circumference and the height by knee height. 

Nutrition-related issues were also recorded, i.e., reduced food intake, loss of appetite, changes in smell and/or taste, nausea, gastrointestinal disturbances, etc. Moreover, diet prescription from the dietetic staff was recorded and analyzed, based on the portion size and the food items provided by the catering service of the hospital. 

In the follow up, the diet prescription during hospitalization was recorded and analyzed according to the hospital food analysis database, in terms of calorie and protein content. Moreover, patients were asked to describe the approximate percentage of the food they consumed (0%, 25%, 50%, 75%, 100%) based on the plate measurement used in the NutritionDay questionnaires [20,21]. Moreover, the reasons for reduced food consumption, the consumption of food items other than those offered in the hospital, the provision of nutritional support (oral nutritional supplements (ONSs), enteral nutrition (EN), parenteral nutrition (PN)), its duration, and the percentage of the dietary needs covered were also recorded. The flow chart of the study is presented in Figure 1. 

The follow up also included information about the nutritional status of the patients, the outcome of the hospitalization, i.e., the LOS, the prevalence of hospital infections (type, time of onset, and outcome), and the cause of death in cases of deceased patients. 

The study was conducted according to the guidelines of the Declaration of Helsinki. All patients provided written informed consent to the inclusion of material pertaining to themselves, acknowledged that they could not be identified via the paper; and that they had been fully anonymized. The study was also approved by the Ethics Committee of the participating hospital.

### 2.3. Statistical Analysis

The sample size was calculated based on the population of the patients with hematological malignancies followed by the hematological clinics in our hospital, (approximately 500 patients/6 months) and the percentage of malnutrition published by other investigators (mean prevalence of malnutrition 23%) [2,3] with a confidence level of 95% and a margin of error of 5%. The power calculation resulted in 166 patients. Descriptive statistics were used for the baseline characteristics of the sample. Normality was explored with the Shapiro–Wilk test. Values are presented as median (interquartile range, IQR) for the non-normally distributed variables and mean ± standard deviation (SD) for the normally distributed ones. Chi-square test was used for correlations between categorical variables, and Spearman non-parametric correlations were performed. The Fisher’s exact test was used for analyses of categories with expected values below five, while the independent-samples Mann–Whitney U test was used for testing between a categorical variable with two levels and non-normally distributed continuous variables and the Kruskal–Wallis H test for categorical variables with more than two levels. IBM SPSS statistics, version 23.0 (IBM Corporation, North Castle, NY, USA) was used for the statistical analysis of the results.

## 3. Results

### 3.1. Study Population, Demographics

The audited population consisted of 247 subjects, 62 of which were excluded for not meeting the inclusion criteria. More specifically, 15 patients refused to participate, 44 were hospitalized for less than 3 days, and 5 were absent during reassessment. Eventually, the final audited population consisted of 185 subjects. The baseline characteristics of the sample are presented in Table 1 and the audit’s results in Table 2. 

### 3.2. Diagnosis Profile of the Patients, Predisposing Factors for DRM

A total of 94 of the 185 patients (50.8%) suffered from lymphoproliferative disorders, 76 (41.1) from myeloproliferative disorders, and 15 (18.1%) from myelodysplastic syndrome and acute leukemias. Ninety-five (50%) had a history of recent chemotherapy administration and five (2.7%) recent radiation therapy. Thirteen patients reported mucositis of different grades and 39 symptoms that could limit their nutritional intake, i.e., dysphagia (14, 7.6%), odynophagia (9, 4.9%) and nausea (16, 8.6%). 

### 3.3. Nutritional Profile and Nutritional Support of the Patients

According to the results, 37.3% of the patients were found to be at high risk of malnutrition upon admission to the hospital. Moreover, 31.9% of them reported reduced food intake in the last six months, and 43 patients (24.3%) reported weight loss greater or equal to 10% of their usual body weight in the last six months. Adding to that, 27.7% of the patients reported reduced or no food intake in the last 5 days, while 70% of them reported normal dietary intake. 

According to nutritional intake during hospitalization, almost 30% of the patients reported reduced nutritional intake. A significant percentage of patients chose to consume food items other than the ones served, namely 59.6% of them. According to the energy provision of the hospital food, the caloric content of the meals provided 1800 (1622–2460) Kcals, while the consumption was significantly lower at 1261.2 (0–2460) Kcals. (Table 2). Despite this difference, only 2.7% (*n* = 5) of the patients were provided with any type of nutritional support. 

For the identification of possible correlations between nutrition and baseline characteristics that could have a significant impact on the outcome of the patients, Spearman two-tailed analysis was performed. Age was negatively correlated with weight loss in the past 6 months (r = −0.208, *p* = 0.005), while it was positively correlated with the MUST score (r = 0.204, *p* = 0.005), and the ECOG performance status (r = 0.267, *p* = 0.000), i.e., the older patients reported higher weight loss in the past six months and a higher risk of being malnourished or having the worst performance status.

Nutritional status was found to be correlated with the performance status of the patients. ECOG performance status was significantly positively correlated with the MUST score (r = 0.293, *p* = 0.000) and negatively correlated with weight loss in the previous six months (r = −0.303, *p* = 0.000), i.e., patients with the worst performance status were at higher risk of being malnourished and vice versa, and reporting higher weight loss in the past six months had a negative impact on performance status. At the same time, nutritional risk as assessed by the use of MUST was positively correlated with the prevalence of hospital infections (r = 0.261, *p* = 0.000). (Table 3).

## 4. Discussion

The early detection and treatment of DRM during hospitalization and the maintenance of a good nutritional status is vital, since undernutrition is associated with an increased risk of hospital infections, delayed wound healing, and a longer hospital stay, as well as increased costs of treatment, and higher morbidity and mortality risk [22,23]. DRM is a significant problem, affecting 20–60% of the hospitalized patients [13,24,25]. Patients with hematologic malignancies are a special group with an increased risk of developing DRM [3,26], as well as a population that has not been studied thoroughly. Through the present audit, we aimed at illuminating the problem of DRM in these patients and exploring the current provision of nutritional care in one of the largest hematologic units in the country. 

Based on our results, detection and subsequent treatment of DRM were found to be inadequate in the audited population. The results of nutritional screening and assessment indicate that 37.3% of the patients were at high risk of DRM, as assessed using MUST. The percentage of patients at high risk of malnutrition, as detected by MUST, was higher than the prevalence of malnutrition reported by other reviewers in patients with hematological malignancies using other screening tools or nutritional assessment [2,3] As screening is used to identify patients at risk of malnutrition with quick and easy measurements, it is common to include patients without established malnutrition but who are at high risk for it, and therefore the percentages are greater than the actual diagnosis. The latter is of great importance, especially after the introduction of the diagnostic criteria of malnutrition by the Global Leadership Initiative on Malnutrition (GLIM), where the initial screening with a validated screening tool has to be confirmed by the estimation of phenotypic and etiological criteria to make the diagnosis [6].

According to the results related to food intake during hospitalization, the fact that almost 60% of the patients chose to eat food items different from the ones served stresses the importance of the improvement of food catering in hospitals. According to the analysis of results from NutritionDay, food consumption during hospitalization is associated with variables related to both patients’ condition and factors related to the quality of hospital food, namely the taste and the smell [27]. Especially in patients receiving therapies with a significant impact on nutritional intake, such as oncological patients, ensuring sufficient intake by optimizing the appearance and the taste of hospital food and providing artificial nutritional support to cover the nutritional deficiencies are of vital importance.

The low number of referrals to dietitians and the provision of nutritional support to only five patients (2.7%) should also be stressed. Even though nutritional screening and the presence of nutrition support teams (NSTs) have been obligatory by law for Greek hospitals since 2012 (Legislation #N4052/2012), the actual percentages of screening on admission are still very low [13]. Moreover, the majority of the hospitals, public and private, have not implemented the law and still do not have functional NSTs. As doctors need to make a referral to the dietetic department before the provision of nutrition consultation, it was noted that only a few referrals were made at the period of the audit. One of the possible reasons for this result could be the lack of sufficient education on clinical nutrition in most of the medical universities in the country. According to published data, medical doctors in Greece demonstrate low knowledge of fundamental principles of clinical nutrition, jeopardizing the provision of high-quality and efficient nutritional support [28]. In the same study, an overestimation of their knowledge was also illuminated, even for doctors that had never received relevant training during their basic studies or as a part of lifelong learning procedures [28]. This situation strengthens the need for medical doctors to enhance their level of knowledge and their awareness of the importance of DRM and at the same time the need for a closer collaboration with specialties trained to provide such services, i.e., clinical nutritionists and dieticians. This need has also been illuminated in the position paper published by the European Society of Clinical Nutrition and Metabolism, where the need for improving the gap of knowledge on nutrition for health care professionals and medical students was addressed [29].

Moreover, the limited personnel in the dietetic departments is another obstacle that limits the effectiveness of the provision of nutritional support. Although according to the current legislation the rate between patients and dieticians should be around 80 to 1, due to the current financial situation and the austerity measures that have been posed in Greece since 2010, the current rate does not exceed 200 to 1 in many of the hospitals in Greece.

To our knowledge, this is the first quality improvement project to address the complicated issue of DRM in patients with hematologic diseases using the corresponding methodology and a nutritional screening tool. Nonetheless, some potential limitations do exist. First, information on the portion of the food consumed by the patients was self-reported using a plate waste model, as the tray collection method was not performed. This may have decreased the validity and reliability of collected data regarding food intake that might include under- or over-reporting. Secondly, this audit is a single-center study; therefore, the results could possibly not be generalized. Finally, the MUST screening tool, although considered valid and reliable in clinical environments, might have low sensitivity, especially in acute care, where the effect of the disease overpasses the effect of BMI. Moreover, in older adults, the cutoff for BMI may be low, predisposing them to established malnutrition even before they are found to be at medium or high risk. 

## 5. Conclusions

In conclusion, DRM was highly prevalent in our population and according to our results, it was not identified; thus, it was poorly treated. Continuous education of medical and nursing students and doctors and a closer interaction between doctors, nurses, and dietitians should be encouraged based on the need for multidisciplinary management of DRM, while the initiation of a multidisciplinary NST should be a priority to address the situation. Improvement of the catering facilities would also improve the quality of the provided food, increasing the acceptability and the overall consumption by patients. Finally, the development and adherence to a local or national plan for the identification and management of DRM will improve the standards of care, and at the same time provide a significant economic benefit for the health care sector.

## Figures and Tables

**Figure 1 diseases-10-00040-f001:**
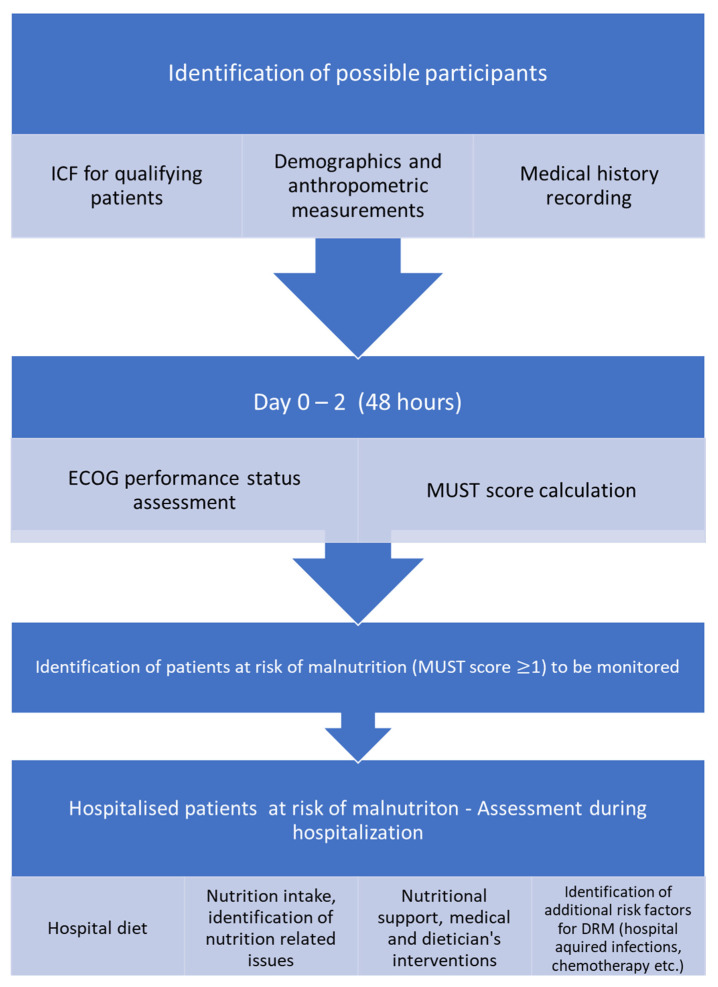
Flow chart of the audit.

**Table 1 diseases-10-00040-t001:** Demographic, anthropometric, and disease characteristics of the audited patients.

Baseline Characteristic	Result
Study Population, Demographics, Anthropometric Data	
Number of patients (*n*)	185
Gender (male)	74
Age, years, median (Interquartile Range (IQR))	62 (25)
ΒΜΙ, Kg/m^2^, median (IQR)	25.38 (4.47)
ECOG, *n* (%)	
0	103 (55.7)
1	48 (25.9)
2	16 (8.6)
3	14 (7.6)
4	4 (2.2)
Diagnoses and predisposing factor for DRM profile	
Diagnosis, *n* (%)	
Lymphoproliferative disorder	94 (50.8)
Myeloproliferative disorder	76 (41.1)
Myelodysplastic syndrome/acute leukemia	15 (8.1)
Dysphagia, *n* (%)	14(7.6)
Odynophagia, *n* (%)	9 (4.9)
Nausea, *n* (%)	16 (8.6)
Recent infection, *n* (%)	11 (11.8)
Recent chemotherapy, *n* (%)	95 (50)
Recent radiation, *n* (%)	5 (2.7)
Recent abdominal surgery, *n* (%)	3 (1.6)
Mucositis, *n* (%)	13 (7)
Grade 1	9 (4.9)
Grade 2	3 (1.6)
Grade 3	1 (0.5)
Head injury, *n* (%)	3 (1.6)
Neurological disease, *n* (%)	4 (2.2)
Nutritional risk and intake	
MUST score on admission	
0, *n* (%)	84 (45.4)
1, *n* (%)	32 (17.3)
≥2, *n* (%)	69 (37.3)
Nutrition in the last 5 days	
Normal, *n* (%)	130 (70.3)
Reduced, *n* (%)	49 (26.5)
Increased, *n* (%)	2 (1.1)
No food intake, *n* (%)	4 (2.2)
Reduced dietary intake in the last 6 months, *n* (%)	59 (31.9)
% Weight loss in the past 6 months	3.13 (9.43)
Serum Albumin on admission (g/dL), Median (IQR)	4.1 (0.70)

BMI: Body Mass Index, ECOG: Eastern Cooperative Oncology Group, MUST: Malnutrition Universal Screening Tool, IQR: interquartile range.

**Table 2 diseases-10-00040-t002:** Results of the audit.

Characteristic	Result
Length of hospital stay (LOS), days (IQR)	10 (15)
Consumption of food other than that provided in the hospital, *n* (%)	
No	43 (23.2)
Sometimes	62 (33.5)
Always	48 (25.9)
No data	15 (8.1)
Calorie content of the meals provided in hospital, median (range)	1800 (1622–2460)
Caloric intake, median (range)	1261.2 (0–2460)
Nutritional support, *n* (%)	5 (2.7)
Hospital-acquired infections	12 (6.5%)

IQR: interquartile range.

**Table 3 diseases-10-00040-t003:** Presentation of the correlations of factors related to nutrition and performance status.

Correlations	r	*p*
Age vs. weight loss in the past 6 months	−0.208	0.005
Age vs. MUST score	0.204	0.005
Age vs. ECOG status	0.267	0.000
MUST score vs. ECOG status	0.293	0.000
ECOG status vs. weight loss in the past six months	0.303	0.000
MUST score vs. hospital infections	0.261	0.000

MUST: Malnutrition Universal Screening Tool, ECOG: Eastern Cooperative Oncology Group.

## Data Availability

Not applicable.

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
