# Peer review of "Are We Identifying Malnutrition in Hospitalized Patients with Hematologic Malignancies? Results from a Quality Clinical Audit"

_diseases, 2022, doi:10.3390/diseases10030040_

Round 1
Reviewer 1 Report
It is an interesting study on a common problem: the lack of identification of patients at nutritional risk and the lack of hospital care regarding nutritional support. Some comments on the manuscript:
Introduction.
The objectives of the study are written at the end of this section: to investigate the prevalence of DRM; level of awareness among the medical team; and the percentage of the patients at risk of malnutrition who receive any type of nutritional intervention. However, in the Material and Methods section authors wrote another objectives (outcomes): To study the possible effect of DRM on hospital infections, mortality and the length of hospitalization (as a secondary outcomes). I suggest writing these outcomes with the other objectives. Also, the outcomes about “the possible effect of DRM on mortality and the length of hospitalization” are not reported, and could be removed.
Material and Methods.
Please briefly indicate the characteristics of the Hematology units. ¿Are they public or private? ¿How many beds do they have? Average hospital admissions per month, for example.
¿In what period of time was the data collected?
It is necessary to describe briefly the MUST screening tool and the ECOG scale.
¿Did you calculate the sample size to meet the objectives?
Please indicate which Ethics and Research Committee approved the protocol and the registration number.
Results.
Page 4, line 157. It is better expressed as a two-tailed rather than a two-sided test (Spearman two-sided).
References.
14 of 25 references (56%) are older than 10 years. I suggest replacing as many as possible with more recent references.
Author Response
We would like to thank you for the thoughtful comments and constructive suggestions, which helped to improve the quality of this manuscript. Please find below a point-by-point response to the reviewer's 1 concerns. We hope that the reviewers will find our responses satisfactory and that the manuscript is now acceptable for publication.
|
Reviewer’s comment |
Response |
|
Reviewer 1 |
|
|
Introduction |
|
|
The objectives of the study are written at the end of this section: to investigate the prevalence of DRM; level of awareness among the medical team; and the percentage of the patients at risk of malnutrition who receive any type of nutritional intervention. However, in the Material and Methods section authors wrote another objectives (outcomes): To study the possible effect of DRM on hospital infections, mortality and the length of hospitalization (as a secondary outcomes). I suggest writing these outcomes with the other objectives. |
The outcomes stated were removed from the material and methods section and transferred to the introduction section. Lines 64-69 “The present study is a cross sectional clinical audit on hospitalized patients with hematologic malignancies aiming to investigate the prevalence of DRM using the Malnutrition Universal Screening Tool (MUST), the level of awareness among the medical team treating this group of patients, the number of referrals to the dietitians, the percentage of the patients at risk of malnutrition who received any type of nutritional intervention and the possible effect of DRM on hospital infections, on length of stay (LoS) and mortality.”
|
|
Also, the outcomes about “the possible effect of DRM on mortality and the length of hospitalization” are not reported, and could be removed |
The outcomes have been removed (Lines 127-131) |
|
Materials and methods |
|
|
Please briefly indicate the characteristics of the Hematology units. ¿Are they public or private? ¿How many beds do they have? Average hospital admissions per month, for example. |
The relevant information has been added. Lines 73 – 75 “The present study was a cross sectional quality clinical audit in two Hematology Units of a tertiary public University Hospital in Athens, Greece. The overall capacity of these units is 70 beds and the average hospital admissions per month is 300” |
|
¿In what period of time was the data collected? |
Relative information has been added. Lines 82-83 “The data collection took part from May 2018 until December 2018” |
|
It is necessary to describe briefly the MUST screening tool and the ECOG scale. |
The MUST score and ECOG scale have been briefly explained Lines 89-96 “The MUST tool is a screening tool introduced by the British Association of Enteral and Parenteral Nutrition (BAPEN), designed to identify patients at risk of malnutrition. [16]. It has been validated in clinical environment, in outpatient clinics, general practice, the community and in care homes [16]. MUST score is calculated by the evaluation of recent unintentional weight loss, Body Mass Index (BMI) and the severity of disease and the total score ranges form 0-2, where 0 corresponds to low risk, 1 to medium risk that needs monitoring and 2 to high risk that necessitates action to treat [16].” And Lines 99-103 “The performance status of the patients was evaluated using the Eastern Cooperative Oncology Group (ECOG) scale, a widely used method to assess the functional status of a patient. It classifies patients to a 5-step scale, with 0 corresponding to fully functional status and 5 to very low performance status and death. [17].” |
|
¿Did you calculate the sample size to meet the objectives? |
The sample size was calculated based on the population of patients with hematologic malignancies followed by our university (n=500), the percentage of malnutrition in this population published by other investigators (mean % = 23%), with a confidence level of 95% and an marginal error of 5%. The power calculation resulted in a sample or 166 patients. Relative information has been added. (Lines 143-147)....... |
|
Please indicate which Ethics and Research Committee approved the protocol and the registration number. |
The protocol was approved by the Hospital Ethics Committee. A relative sentence has been added (Lines 136-137_ |
|
Results |
|
|
Page 4, line 157. It is better expressed as a two-tailed rather than a two-sided test (Spearman two-sided). |
It has been corrected (line 190) |
Reviewer 2 Report
Identifying and treating malnutrition in hospitalized patients 2 with hematologic malignancies. Are we doing enough? Real world results from an audit
Summary: This study by Kanioura et al. aims to describe the prevalence and nutrition support for patients with hematologic malignancies. The overall study is interesting and needed to describe the low level of nutrition support for patients with risk of malnutrition as noted by the authors, although I have some clarifying points.
• The primary objective needs to be clarified – is the outcome awareness or nutritional support? Clinicians may be aware but not take the necessary nutrition care actions. • The MUST tool should be described further in the methodology, including the scoring system and previous validation. • Line 146-147 of the results include interpretations that should be moved into the discussion. • The suggestion that patients at high malnutrition risk are more prone to hospital infections should be further investigated, controlling for other covariates andmalnutrition risk. • In addition to malnutrition risk via the MUST score, the prevalence of malnutrition utilizing anthropometrics should be described in the results and discussion. • Was there any difference in those participants who received nutritional care versus those who did not? • The discussion regarding the consumption of types of food in the hospital is interesting, although meaningful conclusions are unlikely since many different factors can affect these consumption patterns. Recommend focusing on the percentage of foods eaten to indicate whether caloric needs are being met without nutritional support. Estimates of calories consumed are in the table but not discussed in the results or removed from the paper. • The second paragraph of the discussion should be broken into two paragraphs, one detailing the prevalence of malnutrition risk and the other the nutrition support. • There are multiple other limitations of this study that need to be described. Limitations of the MUST scoring need to be described. Additionally, this is a single-center study, and findings may not be able to be generalized.
Author Response
We would like to thank the reviewers for the thoughtful comments and constructive suggestions, which helped to improve the quality of this manuscript. Please find below a point-by-point response to the reviewers’ concerns. We hope that the reviewers will find our responses satisfactory, and that the manuscript is now acceptable for publication.
|
Reviewer 2 |
|
|
The primary objective needs to be clarified – is the outcome awareness or nutritional support? Clinicians may be aware but not take the necessary nutrition care actions. |
Both the doctors’ awareness and the quantitative measurement of the referrals to the dietitians were objectives of the audit. More specifically we indented to investigate the doctors’ awareness of the importance of the early detection of malnutrition and the ways to provide nutritional support in this group of patients. (Lines 82-87) |
|
The MUST tool should be described further in the methodology, including the scoring system and previous validation. |
The MUST tool has been explained, based on the Reviewer 1 comments. Lines 106-113 “The MUST tool is a screening tool introduced by the British Association of Enteral and Parenteral Nutrition (BAPEN), designed to identify patients at risk of malnutrition. [18]. It has been validated in the clinical environment, in outpatient clinics, general practice, the community and in care homes [18]. MUST score is calculated by the evaluation of recent unintentional weight loss, Body Mass Index (BMI) and the severity of disease and the total score ranges from 0 to 2, where 0 corresponds to low risk, 1 to medium risk that needs monitoring and 2 to high risk that necessitates action to treat [18].” |
|
Line 146-147 of the results include interpretations that should be moved into the discussion. |
The interpretation has been deleted |
|
The suggestion that patients at high malnutrition risk are more prone to hospital infections should be further investigated,controlling for other covariates and malnutrition risk. |
The suggestion has been omitted and only the correlation is presented. Lines 240-241 |
|
In addition to malnutrition risk via the MUST score, the prevalence of malnutrition utilizing anthropometrics should be described in the results and discussion. |
Thank you for this comment. We agree that a thorough nutritional assessment would be useful to diagnose malnutrition. As one of the aims of the audit was to investigate the awareness of medical doctors for DRM risk and the No of referrals to dietitians for a full nutritional assessment, the diagnosis of malnutrition or the full nutritional assessment was out of scope of this study. |
|
Was there any difference in those participants who received nutritional care versus those who did not? |
As the number of patients receiving nutritional support was very low (i.e. 5/185) this comparison was not possible. |
|
The discussion regarding the consumption of types of food in the hospital is interesting, although meaningful conclusions are unlikely since many different factors can affect these consumption patterns. Recommend focusing on the percentage of foods eaten to indicate whether caloric needs are being met without nutritional support. Estimates of calories consumed are in the table but not discussed in the results or removed from the paper. |
Thank you for pointing this out. We have added a relevant paragraph as following. Lines 198-203. According to nutritional intake during hospitalization, almost 30% of the patients reported reduced nutritional intake. A significant percentage of patients chose to consume food items other than the ones served, namely 59.6% of them. According to the energy provision of the hospital food, the caloric content of the meals provided 1800 (1622– 2460) Kcals, while the consumption was significantly lower 1261.2 (0-2460) Kcals. (Table 2). Despite this difference, only 2.7% (n=5) of the patients were provided with any type of nutritional support. |
|
The second paragraph of the discussion should be broken into two paragraphs, one detailing the prevalence of malnutrition risk and the other the nutrition support. |
The necessary changes have been made (Lines 253-276) |
|
There are multiple other limitations of this study that need to be described. Limitations of the MUST scoring need to be described. Additionally, this is a single-center study, and findings may not be able to be generalized. |
The limitations proposed have been added (lines 307-317) |
Reviewer 3 Report
Title of study: I feel that you need to change the title of your study and follow PICO method for defining title of your study. When I read your title, it gave be dual feeling either I am going to review a research paper or review paper.
Abstract: Rephrase the sentence "High risk per the MUST score on admission was 37.3% of the audited population".
Remove 6 from 2.16%. Present only one digit after decimal everywhere.
Remove the heading of discussion, and merge the content written for discussion and conclusion in a heading "Conclusion".
Introduction:
I suggest you to make some changes in your introduction section. At first present the definition of DRM, and then connect it with the content of your second paragraph "DRM in hematologic patients is often . . . . . can save approximately $1,000 per patient [12]. and this will be your first paragraph.
In your second paragraph, start from the content present in your current first paragraph: "Novel therapies and improvement in patients’ care have tremendous impact on survival, but despite the advances in cancer research, various studies have indicated high prevalence of disease related malnutrition (DRM) among these patients [2]", then connect this sentence with "Considering the negative impact on" and continued till the objective and rationale of your study.
Methodology:
There are certain things which you need to improve in your methodology section. At first, you need to clearly write which epidemiological study design was adopted. Although, you wrote it is prospective, but it seems more like simple screening and assessment measures like cross-sectional. Secondly, you need to explicitly write your inclusion and exclusion criteria, on what basis did you include and was there any exclusion? Thirdly, write the sampling strategy and sample size. If the sample size was not calculated either write in limitation of your study. Fourthly, add a few headings related to study exposure, and study outcome and covariates. In then explain what was your exposure, what were your outcomes and covairates. Moreover, draw a flow diagram, and indicate different time points when you collected information related to anthropometry, nutrition intervention, MUST and ECOG. You need to specify how many days follow up. For example, at initial visit you did anthropometry call it day 0 or day 1, after 2 days you performed MUST and EGOC, and next day you check ABCD for 3 days . . . A flow diagram will better explain all the test, which you performed. Moreover, once your article will be published, then it will also attract other readers.
I also advice you briefly described MUST and ECOG,N. and its scoring/grading criteria.
Write a separate heading for ethical principals and also explained further about the statistical tests.
Results: I have no major reservation with the quality of your work, but actually the presentation of your work is not appropriate. I suggest you add different headings, such as demographic profile, diagnosis profile: sign & symptoms, nutritional profile and nutrition intervention. Similarly present these headings inside the Table 1.
Also present a figure or table for showing the correlation.
Discussion: Discussion needs a lot of improvement. You need to keep one paragraph for one theme. One for DRM in hematological malagnancies patients, one for EGOC with MUST, EGOC with weight loss, MUST with recurrent infections. Also support the positive and negative association with medical treatment received, with sign & symptoms, and with calorie intake or food consumption during hospitalization. Moreover, add other studies finding for improving the external validity of your study.
Add some recommendations/future direction of your work and also add strength and weakness of your study.
Conclusion: This will automatically improve.
I also suggest you remove the treatment word from title.
Author Response
We would like to thank the reviewers for the thoughtful comments and constructive suggestions, which helped to improve the quality of this manuscript. Please find below a point-by-point response to the reviewers’ concerns. We hope that the reviewers will find our responses satisfactory and that the manuscript is now acceptable for publication.
|
Reviewer 3 |
|
|
Title of study: I feel that you need to change the title of your study and follow PICO method for defining title of your study. When I read your title, it gave be dual feeling either I am going to review a research paper or review paper. |
The title has been changed into “Are we identifying malnutrition in hospitalized patients with hematologic malignancies? Results from a quality clinical n audit |
|
Abstract |
|
|
Rephrase the sentence "High risk per the MUST score on admission was 37.3% of the audited population". |
It has been rephrased Line 23-25 : On admission 37.3% of the audited population were identified at high-risk of malnutrition according to the MUST score |
|
Remove 6 from 2.16%. Present only one digit after decimal everywhere. |
The changes have been made (line 25) |
|
Remove the heading of discussion, and merge the content written for discussion and conclusion in a heading "Conclusion". |
The change has been made (line 27) |
|
Introduction |
|
|
I suggest you to make some changes in your introduction section. At first present the definition of DRM, and then connect it with the content of your second paragraph "DRM in hematologic patients is often . . . . . can save approximately $1,000 per patient [12]. and this will be your first paragraph. |
The suggested changes have been made. Lines 47-52 ….whereas Disease Related Malnutrition (DRM) refers to a complex syndrome that com-bines the detrimental effects of insufficient nutritional intake and the disease-related systemic inflammatory response. [4-6]. DRM in hospitalized adult patients has been connected with increased morbidity, lower performance status, increased length of stay (LoS) and in-creased mortality rates (7) |
|
In your second paragraph, start from the content present in your current first paragraph: "Novel therapies and improvement in patients’ care have tremendous impact on survival, but despite the advances in cancer research, various studies have indicated high prevalence of disease related malnutrition (DRM) among these patients [2]", then connect this sentence with "Considering the negative impact on" and continued till the objective and rationale of your study. |
The introduction has been changed based on the above-mentioned comment. We hope that the above-mentioned insertion resulted in a satisfactory outcome.. |
|
Methodology |
|
|
At first, you need to clearly write which epidemiological study design was adopted. Although, you wrote it is prospective, but it seems more like simple screening and assessment measures like cross-sectional. |
We thank you for your comment. We totally agree and therefore we have amended the audit as a cross sectional one. |
|
Secondly, you need to explicitly write your inclusion and exclusion criteria, on what basis did you include and was there any exclusion? |
Exclusion criteria have been added (line 99-100) |
|
Thirdly, write the sampling strategy and sample size. If the sample size was not calculated either write in limitation of your study. |
The sample size was calculated based on the population of patients with hematologic malignancies followed by our university (n=500/6months), the percentage of malnutrition in this population published by other investigators (mean % = 23%) [2,3], with a confidence level of 95% and an margin ofl error at 5%. The power calculation resulted in 166 patients. Relative information has been added. (Lines 160-164) |
|
Fourthly, add a few headings related to study exposure, and study outcome and covariates. In then explain what was your exposure, what were your outcomes and covairates. |
As the study was an audit of medical practice, it is snapshot of current practice. The exposure was the hospitalization, and the covariates were nutritional intake and the impact of the disease. Relevant headings have been added. |
|
Moreover, draw a flow diagram, and indicate different time points when you collected information related to anthropometry, nutrition intervention, MUST and ECOG. You need to specify how many days follow up. For example, at initial visit you did anthropometry call it day 0 or day 1, after 2 days you performed MUST and EGOC, and next day you check ABCD for 3 days . . . A flow diagram will better explain all the test, which you performed. Moreover, once your article will be published, then it will also attract other readers. |
A flow diagram has been created. (Figure 1) |
|
I also advice you briefly described MUST and ECOG,N. and its scoring/grading criteria. |
The MUST score and ECOG scale have been briefly explained Lines 106-113 “The MUST tool is a screening tool introduced by the British Association of Enteral and Parenteral Nutrition (BAPEN), designed to identify patients at risk of malnutrition. [18]. It has been validated in the clinical environment, in outpatient clinics, general practice, the community and in care homes [18]. MUST score is calculated by the evaluation of recent unintentional weight loss, Body Mass Index (BMI) and the severity of disease and the total score ranges from 0 to 2, where 0 corresponds to low risk, 1 to medium risk that needs monitoring and 2 to high risk that necessitates action to treat [18].” And Lines 118-120 “The performance status of the patients was evaluated using the Eastern Cooperative Oncology Group (ECOG) scale, a widely used method to assess the functional status of a patient. It classifies patients to a 5 scale, with 0 corresponding to a fully functional status and 5 to very low performance status and death. [19].” |
|
Write a separate heading for ethical principals and also explained further about the statistical tests. |
A paragraph on ethical principles has been completed by the phrase “The study was also approved by the Ethics Committee of the participating hospitals. 155-156) The statistical analysis is also expanded, with the power calculation and the more detailed analysis of the statistical tests used for the analysis of the results (Lines 160-164 and 169-173) |
|
Results |
|
|
I have no major reservation with the quality of your work, but actually the presentation of your work is not appropriate. I suggest you add different headings, such as demographic profile, diagnosis profile: sign & symptoms, nutritional profile and nutrition intervention. Similarly present these headings inside the Table 1. |
Thank you for your suggestion. We have added the corresponding headings as requested. These headings have also been incorporated into Table 1. |
|
Also present a figure or table for showing the correlation. |
A table with the statistical significant correlations has been added (table 3) |
|
Discussion needs a lot of improvement. You need to keep one paragraph for one theme. One for DRM in hematological malagnancies patients, one for EGOC with MUST, EGOC with weight loss, MUST with recurrent infections. Also support the positive and negative association with medical treatment received, with sign & symptoms, and with calorie intake or food consumption during hospitalization. |
Discussion has been enriched with two new paragraphs having to do with screening and food consumption. (lines 256-276) |
|
Moreover, add other studies finding for improving the external validity of your study. |
The references have been revised, including more and more recent research papers. |
|
Add some recommendations/future direction of your work and also add strength and weakness of your study. |
Recommendations and future directions have been added (Lines 321-331) |
|
Conclusion: This will automatically improve. |
The conclusion has changed according to the new isnertions in discussion. (Lines 321-331) |
|
I also suggest you remove the treatment word from title. |
The word “treating” has been removed. |